# Effects of plant tissue permeability on invasion and population bottlenecks of a phytopathogen

Gaofei Jiang [1,2,12], Yuling Zhang[2,12], Min Chen[3,12], Josep Ramoneda [4], Liangliang Han[5], Yu Shi[6], Rémi Peyraud [7], Yikui Wang [8], Xiaojun Shi [1], Xinping Chen [1], Wei Ding[1], Alexandre Jousset [2], Yasufumi Hikichi[9], Kouhei Ohnishi[9], Fang-Jie Zhao [2], Yangchun Xu [2], Qirong Shen [2], Francisco Dini-Andreote [10,11], Yong Zhang[1,3] ✉ & Zhong Wei [2] ✉

Pathogen genetic diversity varies in response to environmental changes. However, it remains unclear whether plant barriers to invasion could be considered a genetic bottleneck for phytopathogen populations. Here, we implement a barcoding approach to generate a pool of 90 isogenic and individually barcoded *Ralstonia solanacearum* strains. We used 90 of these strains to inoculate tomato plants with different degrees of physical permeability to invasion (intact roots, wounded roots and xylem inoculation) and quantify the phytopathogen population dynamics during invasion. Our results reveal that the permeability of plant roots impacts the degree of population bottleneck, genetic diversity, and composition of *Ralstonia* populations. We also find that selection is the main driver structuring pathogen populations when barriers to infection are less permeable, i.e., intact roots, the removal of root physical and immune barriers results in the predominance of stochasticity in population assembly. Taken together, our study suggests that plant root permeability constitutes a bottleneck for phytopathogen invasion and genetic diversity.

Understanding the eco-evolutionary processes structuring the diversity of pathogen populations is of key importance to better inform mitigation strategies and develop effective control measures[1]. Genetically diverse pathogen populations generally have more detrimental effects on hosts, faster responsiveness to control measures, and a higher potential to evolve resistance[2]. Population bottlenecks occurring upon pathogen–host interactions directly affect the genetic diversity of establishing pathogen populations[3,4]. These have been found to be linked with host susceptibility and specific infection routes[4,5]. For plant root and stem diseases, phytopathogen

[1]College of Resources and Environment, College of Plant Protection, Interdisciplinary Research Center for Agriculture Green Development in Yangtze River Basin, Southwest University, Chongqing, China. [2]Key Laboratory of Plant Immunity, Jiangsu Provincial Key Laboratory for Organic Solid Waste Utilization, Jiangsu Collaborative Innovation Center for Solid Organic Waste Resource Utilization, National Engineering Research Center for Organic-based Fertilizers, Nanjing Agricultural University, Nanjing, China. [3]College of Environmental Science and Engineering, Shaanxi University of Science & Technology, Xi'an, China. [4]Institute for Research in Environmental Sciences, University of Colorado, Boulder, CO, USA. [5]Department of Biomedical Science, City University of Hong Kong, Kowloon Tong, Hong Kong SAR, China. [6]State Key Laboratory of Crop Stress Adaptation and Improvement, School of Life Sciences, Henan University, Kaifeng, Henan, China. [7]iMEAN, Ramonville Saint Agne, Occitanie, FR, France. [8]Vegetable Research Institute, Guangxi Academy of Agricultural Science, Nanning, China. [9]Faculty of Agriculture and Marine Science, Kochi University, Nankoku, Japan. [10]Department of Plant Science & Huck Institutes of the Life Sciences, The Pennsylvania State University, University Park, PA, USA. [11]The One Health Microbiome Center, Huck Institutes of the Life Sciences, The Pennsylvania State University, University Park, PA, USA. [12]These authors contributed equally: Gaofei Jiang, Yuling Zhang, Min Chen. ✉e-mail: bioyongzhang@swu.edu.cn; weizhong@njau.edu.cn

populations initially undergo biotic and abiotic filtering imposed by local conditions[6]. The invasive pathogens that are able to bypass this initial selection can then initiate the expression of pathogenic genes and start the process of infection[7]. Then, successful invaders are exposed to the plant immune response, which can further reduce the pathogen population size[8]. Together, all these layers of filters can directly impose population bottlenecks that impact the genetic diversity of establishing plant pathogens[9]. However, it has long been a daunting task to track population dynamics and quantify the population and genetic bottlenecks of plant pathogens. As such, this has been limiting our ability to design and optimize control strategies that account more directly for direct aspects of pathogen population diversity and evolution[10].

The effects of natural selection are widely studied and considered to be the main drivers of the success or failure of phytopathogen invasions[11,12], which directly influence their genetic diversity[13]. For instance, the success of pathogen invasions is dependent not only on their capacity to thrive in the rhizosphere but also on their ability to overcome plant root barriers and plant innate immunity[6,8]. This often results in the competitive exclusion[14] or virulence evolution of phytopathogens[15]. However, often overlooked as a mechanism of pathogen evolution, neutral processes can also affect genetic variation in population dynamics[16,17]. This is likely due to the difficulties in quantifying neutral processes in vivo due to the dominance of selective effects[16] and the high genetic/phenotypic variation in phytopathogens[18]. In brief, neutral processes can promote stochasticity in population dynamics via random birth and death and random dispersal of offspring[19]. Collectively, these mechanisms can often explain significant amounts of the variation in population or community structure of microbiomes and impact functionality[20]. In addition, it is also possible that ecological and genetic drift can affect the rate of fixation of specific genes and lineages[13], which could lead to more or less virulent states of the invader population depending on the strength of plant filters to invasion[3,21,22].

The introduction of unique DNA barcodes at neutral chromosomal locations of bacterial individuals offers an opportunity to track bacterial cell lineage dynamics in a high-resolution manner[23–25]. The unique barcodes constitute DNA tags that can be tracked via amplification and sequencing of the genetic sequence where they were initially inserted in the organisms. This method has been applied to investigate bacterial evolution and transmission in the mouse gut[26], resistance evolution to antibiotics[24] and to quantify the relative influence of selection and ecological drift on microbial evolution[27] and community assembly[28]. The method also holds great promise for monitoring fungal plant pathogens[29]. As such, the use and application of this method to isogenic microbial phytopathogen populations can allow for the investigation and quantification of population dynamics provided that there are no fitness differences between barcodes or preexisting phenotypic differences within the barcoded populations[30].

In this work, we use the widespread gram-negative soilborne pathogen *Ralstonia solanacearum* as a model organism. This pathogen is globally recognized as an important phytopathogen accountable for destructive bacterial wilt disease, employing a vast arsenal of pathogenicity determinants to penetrate and colonize plants, including cell wall-degrading enzymes (CWDEs) and type 3 effectors (T3Es)[7]. We investigate a set of eco-evolutionary processes associated with pathogen invasion in planta using a pool of 90 chromosomally tagged isogenic pathogen populations. This approach allows us to track the genetic diversity of the infective populations via amplicon sequencing[24], to quantify population bottlenecks occurring during invasion and to estimate the relative importance of deterministic and stochastic processes structuring *R. solanacearum* populations (Fig. 1). Recent studies have shown that this phytopathogen displays great genetic diversity at both local and global scales[31–33]. This holds true even though most host crops are often homogenous cultivars that

theoretically impose directional selection on pathogen evolution[34]. Here, we specifically test how plant-tissue physical permeability affects the genetic diversity of invading *R. solanacearum* populations. To do so, we experimentally manipulate the permeability of plant roots exposed to phytopathogen populations. We hypothesize that the genetic diversity of the establishing population would be proportional to the permeability of the plant's physical barriers to infection due to population bottleneck effects. To test this hypothesis, we (1) quantify the population bottlenecks and overall genetic diversity of the establishing phytopathogen populations in plants with varying degrees of physical permeability to invasion, (2) determine changes in the lineage composition of the establishing phytopathogen, and (3) measure the relative importance of selection and stochasticity in structuring the assembly of phytopathogen populations as affected by the degree of permeability of plant physical barriers (Fig. 1).

## Results

### Evaluating the potential impact of random DNA barcode introduction on *R. solanacearum* fitness and virulence

We first tested whether our barcoding approach affected *R. solanacearum* fitness and virulence. To this end, we obtained growth curves and measurements of the maximum growth rates of (1) the ancestral *R. solanacearum* OE1-1 wild-type population; (2) each individually barcoded population (total of 90); and (3) the pooled barcoded population. We found no significant differences in the maximum growth rates ($F_{184,91} = 1.072$, $P = 0.343$), biomass ($F_{184,91} = 0.913$, $P = 0.684$), lag phase ($F_{184,91} = 1.048$, $P = 0.389$) or area under the curve ($F_{184,91} = 1.037$, $P = 0.412$) among all tested populations (AONVA, Supplementary Fig. 1), thus showing no effect of the barcodes on strain fitness. We then selected pools of 10, 30, 50, and 90 barcoded populations cultured at equal volumes to verify that we could adjust

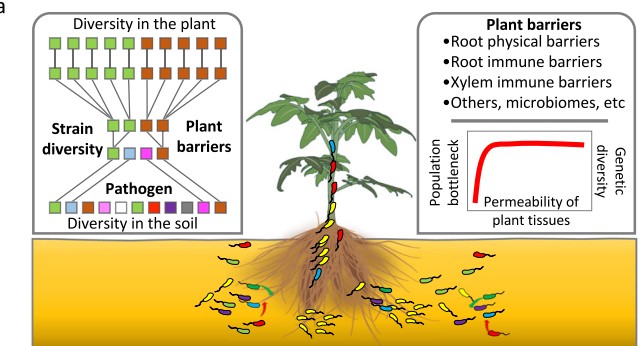

a

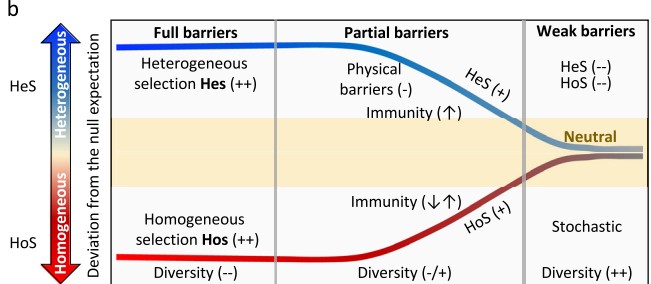

b

**Fig. 1 | Conceptual framework of the processes occurring during phytopathogen invasion. a** Phytopathogen populations in the soil undergo population and genetic bottlenecks due to selection imposed by biotic and abiotic factors. The permeability of plant roots determines the genetic diversity of the established pathogen population inside plant roots. **b** The relative influence of deterministic selection and stochastic assembly processes of pathogen populations during invasion is expected to be associated with the degree of permeability of plant barriers and the genetic diversity of phytopathogen populations. Hos: homogenous selection; Hes: heterogeneous selection; Und: undominated processes.

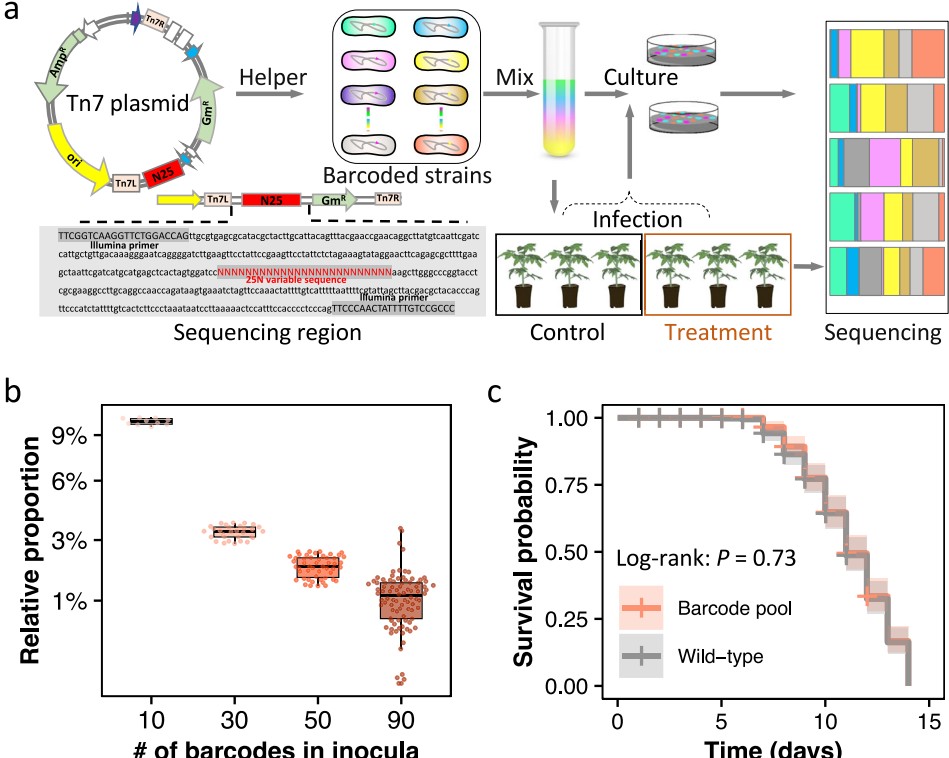

**Fig. 2 | Strain-specific DNA barcoding as an approach to track strain-level population dynamics of phytopathogens. a** General scheme of the random chromosomal barcoding approach and the experimental design. First, we edited a 25-nucleotide random sequence (the 'barcode') and the adjacent marker of selection between the Tn7 arms on the integration plasmid. The helper plasmid and Tn7 plasmid were then integrated at a defined, neutral position in the chromosome of the *R. solanacearum* cells by inducing the transposase machinery. The fate of each barcoded strain can then be tracked by sequencing; **b** proportion of the barcoded individuals after in vitro growth for 24 h initially mixed at equal concentrations with an increasing number of total barcoded strains per inoculum. The *x*-axis represents the starting number, i.e., 10 ($n = 10$), 30 ($n = 30$), 50 ($n = 50$), 90 ($n = 90$), of mixed-barcoded strains. The boxes represent the 25th–75th percentile, lines and dots represent medians and individuals. **c** Survival curve of tomato plants inoculated with the pathogen wild-type strain and the barcoded strain pool of *R. solanacearum*. *P* values > 0.05 indicate no fitness variation of barcoded strains on pathogen virulence based on the log-rank test of survival curves. Source data are provided as a Source data file.

barcode proportions accurately. All barcodes displayed very similar proportions to the initial inoculum (Fig. 2b), which validates our even distributions of the barcoded populations. To test the effect of barcoding on strain virulence, we inoculated pools of the barcoded populations into susceptible tomato plants. This also confirmed that the virulence of barcoded populations had no significant differences from that of the ancestral *R. solanacearum* wild-type strain (log-rank: $P = 0.73$, Fig. 2c and Supplementary Fig. 2). Hence, the results from the control treatments performed are consistent with no significant effects of the barcodes on the expression of key pathogenicity factors, such as CDWEs and T3Es.

### Effect of plant root permeability on population bottlenecks of *R. solanacearum*
To determine the magnitude of the population bottlenecks during *R. solanacearum* invasion, we compared the relative abundance of each individual barcoded strain in the initial 90-barcode pool (Initial) with those collected from the intact roots (IR), wounded roots (WR), and direct xylem inoculation (DXI) infection treatments in the wilt-resistant (R) and wilt-susceptible (S) cultivars. From an initially equal abundance (~1.11%) of each barcoded strain in the inoculum, we found specific barcoded strains that randomly increased to dominance (>75% relative abundance) in seven and four of the 12 replicated S and R plants in the IR treatment, respectively (Fig. 3a, b). Conversely, in the WR treatment, there was no highly abundant barcoded strain in any of the 12 S plants (Fig. 3a), while only one R plant was infected by a barcoded strain at a relative abundance >25% (Fig. 3b). Last, in the DXI treatment, all

barcoded strains were distributed in relatively equal proportions in both S and R plants (Fig. 3a, b).

To determine the impact of plant barrier permeability on pathogen population size, we compared differences in bottleneck size ($N_b$) between the two tomato cultivars and the three infection treatments (Fig. 3c). Both cultivars and infection treatments contributed to variation in bottleneck size with a significant interaction effect (Cultivar: $F_{66,1} = 56.30$, $P = 2.1 \times 10^{-10}$; infection treatment: $F_{66,2} = 79.65$, $P = 2.0 \times 10^{-16}$; interaction: $F_{66,2} = 15.66$, $P = 2.7 \times 10^{-6}$, two-way ANOVA, Fig. 3c). The pathogen populations in the IR treatment had the strongest bottleneck, with $N_b$ values of 4.96 and 1.60 in the R and S plants, respectively (Fig. 3c and Supplementary Fig. 3). The $N_b$ median values in treatments WR and DXI were 10.31 and 28.57 times higher than those of IR in R plants (Resistant: $F_{33,2} = 29.52$, $P = 4.5 \times 10^{-8}$, LSD-test) and 121.75 and 128.65 times higher than those of IR in S plants (Susceptible: $F_{33,2} = 53.04$, $P = 4.9 \times 10^{-11}$, LSD-test, Fig. 3c).

### Impact of plant root permeability on the genetic diversity of *R. solanacearum*
We explored the relationship between plant root permeability and the genetic (i.e., strain-level) diversity of invading *R. solanacearum* populations. We found significant differences in the genetic diversity between infection treatments in both susceptible (ANOVA: $F_{33,2} = 15.66$, $P = 2.0 \times 10^{-16}$, Fig. 4a) and resistant (ANOVA: $F_{33,2} = 63.09$, $P = 5.3 \times 10^{-12}$, Fig. 4b) plants. In all cases, IR had a significantly lower population diversity compared to other infection treatments ($P < 0.001$, LSD-test, Supplementary Fig. 4). No significant differences

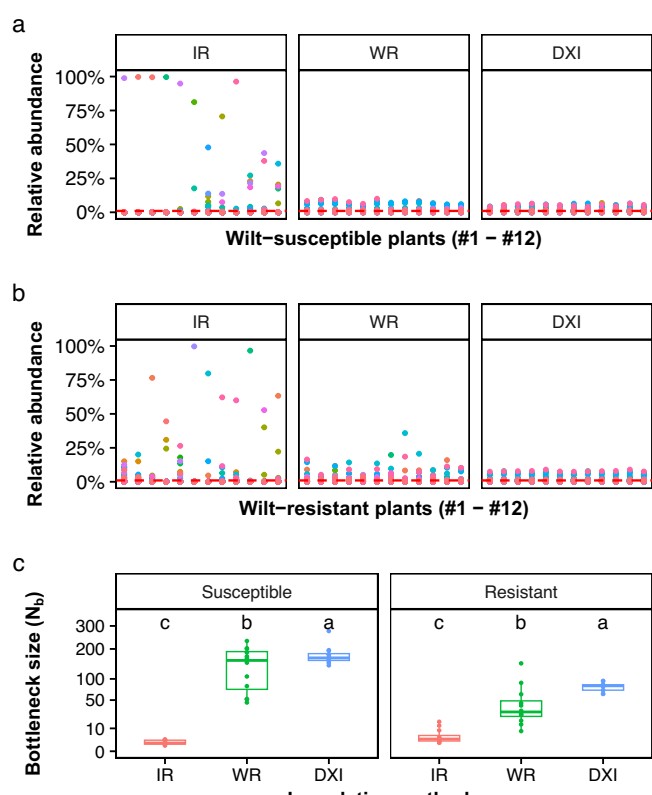

**Fig. 3 | Impact of plant–root permeability on population bottlenecks of *Ralstonia solanacearum*.** Relative abundance of the 90 barcoded strains in the initial inoculum (Initial) in the susceptible (**a**) and resistant (**b**) tomato cultivar. Each colour denotes the identity of the individual barcoded strains. The dashed line shows the average relative abundance (1.11%) of each barcoded strain in the inoculum of the *R. solanacearum* barcoded pool. **c** Comparison of bottleneck sizes ($N_b$) in *R. solanacearum* populations undergoing invasion of wilt-resistant (R) and wilt-susceptible (S) tomato cultivars based on different inoculation methods (i.e., treatments). The treatments are associated with the permeability of the plant's physical barriers to phytopathogen invasion (intact roots, IR; wounded roots, WR; direct xylem inoculation, DXI). Each treatment included 12 repeats ($n = 12$). The boxes represent the interquartile range of the 25th–75th percentile of data, lines and dots represent medians and individuals. Lowercase letters above box plots represent significant differences (ANOVA with LSD test). Source data are provided as a Source data file.

in genetic diversity were found between WR and DXI (Shannon index: $P = 0.08$; Richness: $P = 0.54$; Evenness: $P = 0.13$, LSD-test, Fig. 4a, b and Supplementary Fig. 4). Importantly, the standard deviation reflecting the variation in diversity within each treatment was approximately 2 and >7 times higher in the IR treatment than in the WR and DXI treatments, respectively (Fig. 4a, b and Supplementary Fig. 4). Strain diversity and population bottlenecks covaried strongly in a range of $N_b = 0$–50, as observed through the comparison of the $N_b$ values and *R. solanacearum* alpha diversity metrics. Genetic diversity significantly increased with increasing $N_b$ value (Shannon index: $\rho = 0.97$, $P = 2.2 \times 10^{-16}$, $N_{bp} = 14.97$; richness: $\rho = 0.87$, $P = 2.2 \times 10^{-16}$, $N_{bp} = 70.69$; evenness: $\rho = 0.95$, $P = 2.2 \times 10^{-16}$, $N_{bp} = 10.36$, Fig. 4c and Supplementary Fig. 5a, b). This relationship between strain diversity (Shannon index, richness, and evenness) and population bottleneck was not affected by host type (Supplementary Fig. 5c–e).

### Plant–root permeability impacts phytopathogen lineage composition
The composition of the invading phytopathogen populations was influenced by both host type and infection treatment (PERMANOVA:

$R^2 = 0.03$, $F_{66,1} = 2.93$, $P = 0.014$ for host type; $R^2 = 0.32$, $F_{66,2} = 16.91$, $P = 2.0 \times 10^{-16}$ for infection treatment; $R^2 = 0.03$, $F_{66,2} = 1.81$, $P = 0.033$ for interaction effect, Supplementary Fig. 6). The variability of invading phytopathogen populations differed across infection treatments in S (PERMDISP: $F_{33,2} = 2676.6$, $P = 0.001$, Fig. 4d) and R (PERMDISP: $F_{33,2} = 196.29$, $P = 0.001$, Fig. 4e and Supplementary Fig. 6a, b). The compositional dispersion of *R. solanacearum* populations within plant roots was influenced by the host type and the infection treatment with a significant interaction effect (Cultivar: $F_{66,1} = 7.07$, $P = 0.010$; infection treatment: $F_{66,2} = 734.94$, $P = 2.0 \times 10^{-16}$; interaction: $F_{66,2} = 20.27$, $P = 2.0 \times 10^{-16}$, two-way ANOVA). These effects segregated small population sizes and low genetic diversity in the IR treatment from other treatments ($P = 1.4 \times 10^{-7}$, LSD-test, Supplementary Fig. 6c). The significant correlation between the $N_b$ value and lineage dispersion further confirmed this finding ($\rho = -0.74$, $P = 6.3 \times 10^{-16}$, $N_{bp} = 46.09$, Fig. 4f and Supplementary Fig. 6d).

### Plant–root permeability impacts the assembly processes structuring *R. solanacearum* populations
To determine the relative influence of assembly processes structuring *R. solanacearum* populations during invasion, we used the beta artificial taxon index (βATI) and the Bray–Curtis-based Raup-Crick metric ($RC_{bray}$) metrics (Fig. 5) and the goodness of fit to Sloan's neutral model (Supplementary Fig. 7). First, we found invading populations of *R. solanacearum* to be mostly structured by deterministic selection processes in treatment IR (reflecting a low permeability to invasion– and associated with greater population and genetic bottlenecks). In contrast, the relative influence of stochastic processes increased in treatments WR and DXI (reflecting a greater permeability to invasion and associated with lower population and genetic bottlenecks) (Fig. 5). In detail, the IR treatment resulted in a βATI > 2 (Fig. 5a), which suggests the predominance of variable selection, both in R and S plants (Fig. 5b). At the intermediate level of permeability (treatment WR), the proportion of variable selection was found to be lower, and homogeneous selection (βATI < 2) was the predominant assembly process. Last, when plant barriers were highly permeable to invasion (treatment DXI), the population assembly was mostly structured by stochastic processes (|βATI|<2), further determined via $RC_{bray}$ analysis as undominated processes, in both S and R plants (Fig. 5b). Based on Sloan's neutral model, the frequency of barcoded strains in the IR treatment (susceptible: $R^2 = 0.07$, resistant: $R^2 = 0.01$) showed a poorer fit to the model compared to the WR (susceptible: $R^2 = 0.53$, resistant: $R^2 = 0.49$) and DXI (susceptible: $R^2 = 0.88$, resistant: $R^2 = 0.81$) treatments (Supplementary Fig. 7a). The population bottlenecks ($N_b$) had a strong correlation with βATI values, whereby a reduction in the strength of the population bottleneck during invasion shifted the assembly processes from predominantly deterministic to predominantly stochastic based on the established pathogen invader populations (Fig. 5c). The goodness of fit to Sloan's model ($R^2$) increased with bottleneck size and plant root permeability, suggesting an increasing role of stochasticity in structuring the assembly of the phytopathogen populations (Supplementary Fig. 7c).

## Discussion
Understanding the processes and mechanisms associated with pathogen invasions can inform strategies for the prevention and mitigation of their impacts[30]. In this study, we provided a comprehensive analysis of the impacts of plant physical barrier permeability on the population size, genetic diversity, and assembly processes structuring phytopathogen populations inside plant roots upon invasion. In particular, we showed that the application of the chromosomal insertion of random DNA barcodes is a powerful tool to track strain level distributions in planta. This tool can be further used to quantify multiple parameters of pathogen fitness and evolution, including strain specificity and population and genetic bottlenecks.

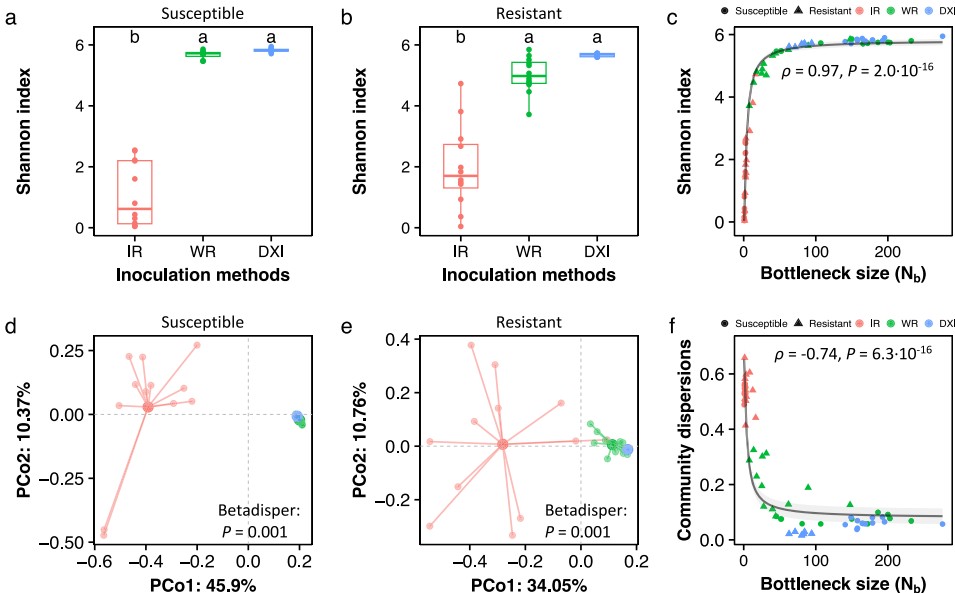

**Fig. 4 | The genetic diversity of *Ralstonia solanacearum*, cell lineage composition, and their correlation with population bottlenecks are affected by plant root permeability.** Genetic diversity (Shannon index) of *R. solanacearum* populations infecting susceptible (**a**) and resistant (**b**) tomato cultivars across the treatment of intact roots (IR), wounded roots (WR), and direct xylem inoculation (DXI). Each treatment included 12 repeats ($n = 12$). Lowercase letters above box plots represent significant differences (ANOVA with LSD test). The boxes represent the interquartile range of the 25th–75th percentile of data, lines and dots represent medians and individuals. **c** Correlation between bottleneck size ($N_b$) and the genetic diversity of invasive *R. solanacearum* populations. Principal coordinates analysis (PCOA) based on Bray–Curtis distances displays the differences in *R. solanacearum* lineage composition across inoculation treatments in resistant (**d**) and susceptible (**e**) plants by permutation multivariate analysis of dispersion test (PERMDISP). **f** Correlation between *R. solanacearum* population bottleneck size ($N_b$) and community dispersion after invasion. In (**c**) and (**f**), $\rho$ represents Spearman's rank correlation coefficient. Lines and intersections represent the mean and 95% CI of the fitting curve of the asymptotic regression model. Source data are provided as a Source data file.

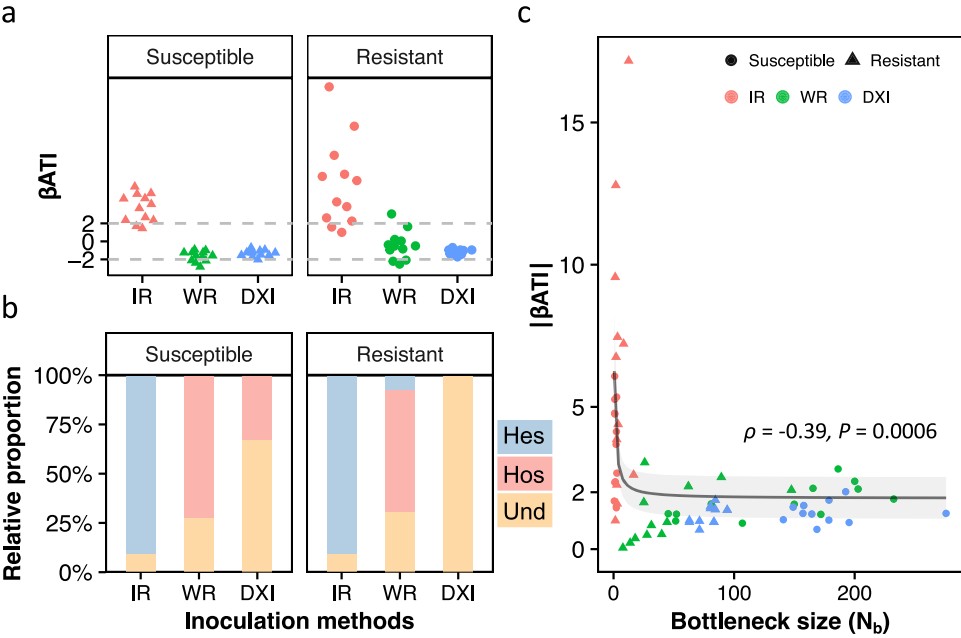

**Fig. 5 | Effects of plant–root permeability on the ecological processes structuring the assembly of *R. solanacearum* populations. a** The value of the beta artificial taxon index (βATI) of resistant and susceptible cultivars in treatments IR (intact roots), WR (wounded roots), and DXI (direct xylem inoculation) representing distinct levels of permeability to pathogen invasion; **b** relative contribution of assembly processes structuring the phytopathogen populations in planta; Hos: homogenous selection; Hes: heterogeneous selection; Und: undominated processes; **c** relationships between βATI values and the population bottleneck ($N_b$) of the pathogen populations during invasion. $\rho$ represents Spearman's rank correlation coefficient; Line and intersection represent the mean and 95% CI of the fitting curve of the asymptotic regression model. Source data are provided as a Source data file.

We validated that the use of this random barcoding system did not impact *R. solanacearum* fitness between populations, providing an unbiased system to quantify patterns of invasive pathogen population dynamics in vivo. The assumption of equal fitness has traditionally made the quantification of population dynamics very challenging in both natural and experimental systems[35,36]. For example, the use of random barcode tracking of *E. coli* lineages exposed to subinhibitory antibiotic concentrations revealed that preexisting mutations can be responsible for the fixation of particular lineages rather than adaptive mutations[24]. We acknowledge that in our study, however, we cannot rule out whether preexisting mutations could have influenced the estimates of population dynamics across our experimental treatments. Most importantly, *R. solanacearum* used in this study would not favour spontaneous mutations endorsed by a quantified mutation rate[37] and the timeframe needed for adaptive mutations (previously reported to only be after 13 in planta serial passages[15]). Furthermore, our barcode lineage composition data indicate that in the treatment where selection dominated (intact roots, IR), the random barcode composition was the most variable, thus supporting the fact that prior adaptive mutations were not an important factor accounting for the magnitude of selection across our treatments.

The permeability of plant physical barriers was found to impose population and genetic bottlenecks on pathogen invasive populations in planta. A higher permeability of physical barriers to infection allowed for a larger diversity of invasive populations to establish in plant roots. Population bottlenecks constitute an important driver of pathogen evolution, whereby stronger bottlenecks reduce the changes for subsequent adaptive evolution for pathogenicity and resistance[38]. The genetic diversity of the invaders was also greater when the permeability of plant physical barriers was high but did not linearly vary with bottleneck population size. Specifically, only in a range of small population sizes resulting from strong population bottlenecks ($N_b$ value < 50) was the population size linearly correlated with the genetic diversity of the pathogen. As such, the genetic diversity remained constant within the range of population bottleneck sizes of 50–200 ($N_b$ value) (Fig. 4c). This occurred because most of the original populations were preserved even at intermediate levels of barrier permeability, as observed in the wounded root (WR) treatment. This finding has two important implications. First, susceptible plants due to wounding or immune suppression can act as hotspots for the genetic diversity of pathogens even when these pathogens undergo population bottlenecks. This can explain the levels of genetic diversity reported for *R. solanacearum* both at small[33] and large spatial scales[31,39]. Second, given the importance of genetic diversity for pathogen evolution[2], our results indicate that control and mitigation measures cannot rely solely on diminishing population sizes to prevent the evolution of resistance. In fact, we found evidence to support that even with population bottlenecks, the genetic diversity of the pathogen in vivo can be preserved in physically wounded and immune-suppressive individuals.

The pathogen populations under intact root barriers to invasion were highly variable, while both partially and completely permeable root barriers led to all lineages being preserved. This also resulted in a predictable composition of phytopathogen populations in planta. In light of host-microbe evolution, this indicates that infection resistance can impose random bottlenecks that determine the genetic composition of the infective population and not necessarily select for new strain lineages[40]. For example, rapid evolution is reported as an important mechanism of genetic variation in phytopathogens that is linked to high genetic variation within infective pathogen populations[41]. Studies have shown that *R. solanacearum* can evolve to increase fitness[15] and enhance xylem colonization[42] following a rewiring of virulence gene networks after invasion. While we did not observe obvious fitness differences across our tested lineages that would support rapid evolution, we noted that it is difficult to determine whether particular lineages become dominant due to stochasticity or phenotypic changes. However, it is important to stress that in instances where particular microbial lineages are deemed to have evolved resistance, this might not be the case but a result of random drift, which is prevalent once the pathogen reaches the xylem, as we showed in the direct xylem inoculation (DXI) treatment.

The least permeable barrier to invasion (IR) resulted in a population assembly in planta to be dominated by variable selection, whereas the lack of a barrier to invasion (DXI) led to a stochastic population assembly. In fact, given the larger bottleneck size of pathogen invasion in the absence of a plant barrier (DXI), it is plausible to expect stochasticity to be a dominant process. Under normal conditions, the pathogen faces diverse barriers to infection upon contact with the root surface, including the rhizosphere microbiome[6,43], physical entry into the cortex, plant immune response, and physical transport into the xylem[4]. These filters underlie variable selective forces that cause strong population and genetic bottlenecks that determine the size and diversity of the infective populations. The strength of selective forces may vary[44,45], depending on strain-specific pathogenic determinants, such as CDWEs and T3Es, which allow the dismantling of physical and immune barriers, respectively[7]. The physical and immune barriers to *R. solanacearum* infection can decrease the effective population size by 97.9% and 99.8% in tomato plants, respectively[46]. Once small founder populations establish and proliferate within the xylem by *R. solanacearum* infections[7,46], their dynamics are mostly structured by stochastic processes[47]. Under these conditions, populations that underwent strong bottlenecks are more likely to contain lineages that unpredictably drift to fixation or to extinction[48,49]. It is also worth acknowledging the limitation of the beta nearest taxon index (βNTI) due to the artificial nature of the barcodes. In brief, the random assignment of barcodes does not align with the assumption of phylogenetic conservatism (often used in βNTI studies)[49]. As such, we used a beta artificial taxon index (βATI) to avoid confusion. Most importantly, our βATI results corroborate the output obtained from Sloan's model (a null model that does not include phylogenetic information)[50].

In this study, we investigated the effects of plant barriers to invasion on pathogen population bottlenecks, genetic diversity, and the interplay of stochastic and deterministic processes structuring pathogen populations upon invasion. We implemented a random chromosomal barcoding system that can be extended to studies focusing on the ecology and evolution of plant-associated microbial populations. This methodology can be used to track strain-level population dynamics over time, for example, in response to environmental disturbances or across plant growth stages[49]. When used on populations with known preexisting mutations, random barcoding can aid the testing of the fitness of pathogens or microbial inoculants (e.g., strain-level variation of host-associated microorganisms).

Rapid evolution is gaining traction as an important driver of pathogen resistance in multiple cropping systems[51]. The use of random chromosomal barcoding opens new opportunities for prospective experimental designs to investigate the importance of biological and abiotic factors controlling the fixation of particular genes and strains during infection and proliferation in plant roots and then distinguish those effects from mechanisms associated with rapid evolution. A more diverse manipulation of barriers to invasion can also enhance our understanding of the specific factors contributing to population bottlenecks during infection[4]. Taken together, our study provides a new perspective on strain-level phytopathogen population dynamics mediated by the permeability of plant physical barriers with direct implications for the establishment, fixation, and evolution of plant–pathogen interactions.

## Methods

### Design of a Tn7-based random barcoding system and chromosomal barcoding of *Ralstonia solanacearum*

The wild-type *Ralstonia solanacearum* strain OE1-1 was barcoded with the Tn7-based chromosomal integration system[52]. Unique random barcodes were integrated into a neutral region of the chromosome of *R. solanacearum*. Specifically, a total of 90 random 25-bp barcodes (Fig. 1a; Supplementary Data 1) were designed using the *Biostrings* R package v2.64.1[53] and cloned and inserted into the pUC18-miniTn7T-Gm delivery plasmid at the restriction sites of *BamH* I (TAKARA, Dalian China) and *Hind* III (TAKARA, Dalian China)[52], resulting in the pUC18-barcode plasmid. By coelectroporation with the helper plasmid pTNS2[54], the barcode-carrying cassette was integrated into the 25-bp region downstream (neutral region) of the *glms* gene in the *R. solanacearum* chromosome via the miniTn7 transposition machinery[52]. A total of 90 uniquely barcoded isogenic *R. solanacearum* individuals were cultivated in BG medium (10 g.L$^{-1}$ bacto peptone, 1 g.L$^{-1}$ casamino Acids, 1 g.L$^{-1}$ yeast extract and 5 g.L$^{-1}$ glucose) amended with 25 µg.mL$^{-1}$ gentamycin overnight at 28 °C. The population density was adjusted to an optical density of 1.0 at 600 nm (OD$_{600}$). The adjusted suspensions of barcoded *R. solanacearum* strains were mixed at equal volumes to obtain a pooled mixture with equal sizes of the barcoded strains. The library was distributed into aliquots and stored at −80 °C (see Fig. 1a).

We verified that transposition was successful by selectively plating the strains in BG medium containing gentamycin and subsequently performing PCR and sequencing of the barcode region. To do so, an aliquot of the pooled library was grown overnight at 28 °C in BG medium containing 25 µg.ml$^{-1}$ gentamycin, and the genomic DNA was extracted using the MiniBEST Bacteria Genomic DNA Extraction Kit (TAKARA, Dalian China). DNA fragments containing the barcode tags (ca. 200-bp) were PCR amplified with the primer set ITS-L: 5′-TTCGGTCAAGGTTCTGGACCAG-3′ and ITS-R: 5′-GGGCGGACAAAA-TAGTTGGGAA-3′ and sequenced on an Illumina MiSeq platform (Illumina, Inc., San Diego, California, USA). The barcode sequences and primers used in this study are shown in Supplementary Data 1.

### Analysing the growth of *Ralstonia solanacearum* populations barcoded with random DNA sequences

To determine whether random DNA barcodes imposed variable fitness effects across the *R. solanacearum* populations, we determined the growth curves of each of the 90 uniquely barcoded *R. solanacearum* populations, as well as of the mixed population pool. All barcoded *R. solanacearum* populations were grown in BG medium supplemented with 25 µg.ml$^{-1}$ gentamicin at 28 °C overnight, washed in 0.8% (*w/v*) saline solution, and adjusted to an OD$_{600}$ of 1.0 using ddH$_2$O. We then inoculated 2 µl of each strain into 96-well cell culture plates containing 198 µl of minimal medium ($1.25 \times 10^{-4}$ g. L$^{-1}$ FeSO$_4$·7H$_2$O, 0.5 g. L$^{-1}$ (NH$_4$)$_2$SO$_4$, 0.05 g. L$^{-1}$ MgSO$_4$·7H$_2$O, 3.4 g. L$^{-1}$ KH$_2$PO$_4$ and supplemented with 20 mM L-glutamate as the sole carbon source) per well and cultured the inoculate at 28 °C for 48 h with shaking at 170 rpm. Cell growth was measured using OD$_{600}$ at 4 h intervals with a SpectraMax M5 Plate reader (Molecular Devices, Sunnyvale, CA, USA). The maximum growth rate and biomass, lag-phase time and area under the curve were determined using the *gcFitModel* function to fit growth data with the *grofit* R package v1.1.1[55].

### Experimental design of *R. solanacearum* invasion and pathogenicity in tomato plants

We designed a full factorial experiment using a wilt-susceptible (Ailsa Craig) and a wilt-resistant (Hawaii 7996) tomato cultivar. These cultivars were subjected to pathogen inoculation based on three predefined invasion categories that reflect decreasing levels of permeability of plant physical barriers to invasion: (1) intact roots (IR)−4-week-old seedlings were soaked in a solution containing the pathogen pool at a concentration of $10^7$ cfu.g$^{-1}$ soil; (2) wounded roots (WR)−root wounds were artificially caused by performing two cuts at the meristic sides of the plant roots, followed by pathogen pool inoculation via root soaking; and (3) direct xylem inoculation (DXI)−direct pathogen inoculation into the xylem using 2 µl of cell suspension (population pool) at $10^8$ cfu.ml$^{-1}$ into the freshly cut surface of tomato petioles. This experimental design (2 cultivars × 3 invasion methods × 12 replicates) was used to evaluate the extent to which decreasing levels of permeability reflect reduced magnitudes of the selective filters to pathogen invasion (no barrier removal, IR; external epidermis barrier removal by artificial root wounds, WR; and total barrier removal via direct xylem inoculation, DXI)[46]. Plants were grown in a greenhouse under controlled conditions (temperature of 30 °C, relative air humidity of 80%, and 14 h of light) and watered every two days at the time of disease incidence recording[56].

Tomato tillers were excised at -10 cm in length when wilting symptoms appeared. The tillers were weighed, cut into 1 cm pieces, and dipped in distilled water for 20 min to release *R. solanacearum* cells. Pathogen cell densities were determined using serial dilution and plating. In addition, *R. solanacearum* cells were harvested by centrifugation, genomic DNA was extracted, and samples were further subjected to amplicon sequencing. It is important to note that the *R. solanacearum* wild-type and the barcoded pool cannot cause wilt disease in the resistant cultivar (Hawaii 7996). However, the pathogen can still proliferate inside plant roots of the resistant cultivar. As such, pathogen cells were harvested in resistant plants from the stems 21 days postinoculation.

### DNA extraction and sequencing

The DNA of the invasive *R. solanacearum* populations collected from plant stems was extracted using the MiniBEST Bacteria Genomic DNA Extraction Kit Ver.3.0. (Takara Co., Ltd., Tokyo, Japan), following the manufacturer's protocol. DNA quality and quantity were determined using a NanoDrop 1000 system (Thermo Scientific, USA) prior to PCR amplification of the barcode-carrying cassette. The PCR conditions were as follows: 95 °C for 5 min, followed by 30 cycles at 95 °C for 30 s, 55 °C for 30 s, and 72 °C for 45 s, with a final extension at 72 °C for 10 min. We used the primer set ITS-L/ITS-R linked to the complementary sequence of Illumina adaptor primers for sample demultiplexing in a second PCR. Indexing PCRs were performed in 20 µl reactions containing 4 µl of 5 × FastPfu Buffer, 2 µl of 2.5 mM dNTPs, 0.8 µl of pUC-L (5 µM), 0.8 µL of ITS-R (5 µM), 0.4 µl of FastPfu Polymerase, and 10 ng of template DNA, replenished with deionized sterile water to 20 µl. Amplicons were extracted from 2% agarose gels and purified using the AxyPrep DNA Gel Extraction Kit (Axygen Biosciences, Union City, CA, USA). Sequencing was performed on an Illumina MiSeq platform (Illumina, Inc. San Diego, California, USA).

### Sequence processing

Raw sequences were processed using the UPARSE pipeline[57]. In brief, read pairs of each sample were assembled, and sequences were screened with the following criteria: (1) 250 bp reads with an average quality score <20 over a 10 bp sliding window were truncated, and reads shorter than 50 bp were removed; (2) reads containing ambiguous characters within the barcode region were also removed; and (3) only paired ends that overlapped more than 10 bp were assembled, and all singletons were discarded. Quality-filtered sequences were clustered into operational taxonomic units (OTUs) at 97% nucleotide identity using the Deblur algorithm[58]. Sequences were then analysed using *uclust*[59] against our internal database of the 90 barcode tags based on 100% sequence identity. This procedure was carried out to cluster the reads within each specific barcoded population.

## Statistical analyses

**Estimation of population bottleneck.** The population bottleneck of the pathogen during invasion was calculated as previously reported[23]. The mathematical models developed in population genetics, specifically the estimation of the effective population size ($N_e$) based on the temporal allele frequency of barcodes, were used to estimate the bottleneck size ($N_b$) of the *R. solanacearum* invader population. To estimate the $N_b$ values, it is assumed that changes in the barcode frequencies within plants are introduced by the survival of *R. solanacearum* cells passing through invasion barriers. This can be modelled as:

$$\hat{F} = \frac{1}{k} \sum_{i=1}^{k} \frac{(f_{i,s} - f_{i,0})^2}{f_{i,0}(1 - f_{i,0})} \tag{1}$$

and

$$N_b \approx N_e = \frac{g}{\hat{F} - \frac{1}{S_n} - \frac{1}{S_s}}, \tag{2}$$

where $k$ represents the total number of distinct barcodes, $f_{i,0}$ is the frequency of barcode $i$ at time 0, $f_{i,s}$ is the frequency of barcode $i$ at sampling, $g$ is the number of generations during competitive growth, and $S_n$ and $S_s$ are the sample sizes used to determine the population composition at time 0 or at sampling. The frequency data of each barcode for this calculation are shown in Supplementary Data 2.

**Statistical analyses.** Statistical analyses were performed in R v4.1.1[60]. We used the *vegan* package v2.6-2[61] to estimate alpha diversity indices (i.e., observed richness, Shannon index, and Pielou evenness). Analysis of variance (ANOVA) was used to determine the statistical significance of cultivars and infection treatments and their interaction effect using the package *agricolae* v1.3-5[62]. Principal coordinate analysis (PCoA) based on Bray–Curtis distances was carried out using the *pcoa* function in the *ape* package v5.6-2[63]. Permutational multivariate analysis of variance (PERMANOVA) was used to test for significant differences in inoculation treatments using the *adonis* function in the *vegan* package v2.6-2[61]. Permutation multivariate analysis of dispersion (PERMDISP) was used to test for differences in multivariate dispersion of *R. solanacearum* barcoded strains within each inoculation treatment using the *betadisper* function in the *vegan* package v2.6-2[61]. Correlations between diversity indices and the *R. solanacearum* dispersion axes to $N_b$ values were performed using Spearman's rank correlations implemented in the *cor.test* function of *ggpubr* package v0.4.0[64]. Curves were fitted with an asymptotic regression model (concave for the diversity indices and convex for dispersion) in response to the exponential of reciprocal $N_b$ values using the *lm* function in the *stats* package v4.1.1[60]. The $N_b$ breakpoint ($N_{bp}$) of fitting curves was fixed using the *segmented* function in the *segmented* package v1.6-4[65].

**Analysis of ecological processes structuring population assembly.** We used two distinct models to investigate the effects of plant root permeability on the ecological processes structuring phytopathogen populations in planta. First, the assembly processes structuring *R. solanacearum* barcoded populations were quantified using the beta nearest taxon index (βNTI) and the Bray–Curtis-based Raup-Crick metric (RC_bray)[66,67]. By considering the differently barcoded populations as artificial 'species' and the 25 bp barcodes as the 'variable regions of a housekeeping gene', we were able to determine the relative importance of stochastic and deterministic processes structuring *R. solanacearum* populations in response to plant root permeability to invasion (Fig. 1b). We first generated a maximum likelihood phylogenetic tree based on a multiple sequence alignment of distinct barcode DNAs using the parallel perturbed probcon method in MUSCLE v5[68].

We then calculated the beta artificial taxon index (βATI) adapted to the βNTI metric under the "taxa.lables" null model with 999 permutations using the *ses. MNTD* and *comdistnt* functions in the *picante* package v1.8.1[69]. Specifically, |βATI| > 2 indicates the predominance of deterministic processes, where βATI > 2 and βATI < −2 indicate heterogeneous and homogeneous selection, respectively[20]. When |βATI| values were <2, RC_bray was used to partition the relative influence of distinct stochastic processes: RC_bray < −0.95, RC_bray > 0.95, and |RC_bray| < 0.95 indicate the predominance of homogenizing dispersal, dispersal limitation, and undominated processes (drift), respectively[66]. We also used Sloan's model[50] to estimate expectations in abundances and frequencies of *R. solanacearum* barcoded strains under the assumption of stochastic immigration, birth and death (i.e., neutral population dynamics). In brief, the Sloan model estimates the immigration rate ($m$) to the in planta populations by comparing the barcode frequency across plants. The barcode distribution was fitted to the Sloan model using the *neutral.fit* function in the *MicEco* package v0.9.19[70]. Calculations of 95% confidence intervals for the Sloan model were used to detect whether the frequency of barcoded strains was above or below the data distribution expectation. The parameter $R^2$ indicates the goodness of fit of the model.

## Reporting summary

Further information on research design is available in the Nature Portfolio Reporting Summary linked to this article.

## Data availability

All data underlying this study are available in RsBarcode [https://doi.org/10.5281/zenodo.7807998] at Zenodo, and all raw sequencing data are deposited at the National Genomics Data Center of China (ID: PRJCA016125). Source data are provided with this paper.

## Code availability

All codes used for data analyses and figure visualization in this work are available in RsBarcode [https://doi.org/10.5281/zenodo.7807998] at Zenodo.

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

## Acknowledgements

We thank Chen Liu and Zelong Zhao for their fruitful discussions and suggestions. This research was financially supported by the National Natural Science Foundation of China [42090062 (G.J. & F.J.Z.), 42090064 (Q.S.), 42325704 (ZW), 42277113 (Z.W.), 32170180 (Y.Z.) and 42007038 (G.J.)], the Fundamental Research Funds for the Central Universities [KYT2023001 (Z.W.), XUEKEN2023039 (Q.S.), and KYCXJC2023007 (G.J.)], the Natural Science Foundation of Jiangsu Province (BK20230102 to G.J.), the Swiss National Science Foundation (Early Postdoc Mobility P2EZP3_199849 to J.R.) and China National Tobacco Corporation [110202101047(LS-07) to G.J.]. the Jiangsu Agricultural Science and Technology Innovation Fund [CX(22)1004 to Y.X.] and the Jiangsu Carbon Peak & Carbon Neutrality Science and Technology Innovation Special Fund (BE2022423 to Q.S.), the USDA National Institute of Food and Agriculture and Hatch Appropriations under Project PEN04908 (7006279 to F.D.-A.).

## Author contributions

Author contributions following the CRediT taxonomy (https://credit.niso.org) are as follows: Conceptualization: G.J., Y.Z. and Z.W.; Resources: G.J., Y.Z., and Z.W.; methodology: G.J., Y.Z. and A.J.; data curation: G.J., Y.L.Z., M.C., L.H. and Y.Z.; formal analysis: G.J., Y.L.Z., M.C., Z.Y., Y.S. and R.P.; funding acquisition: G.J., J.R., Z.W., Y.Z., W.D., F.J.Z., Y.C. and Q.S.; investigation: G.J., Y.L.Z., M.C., L.H. and Y.Z.; project administration: G.J., Z.W., Y.Z. and Q.S.; supervision: G.J., Z.W. and Y.Z.; software: G.J. and Y.L.Z.; visualization: G.J. and Y.L.Z.; writing—original draft: G.J., J.R., Y.L.Z., Z.W. and F.D.A.; writing—review & editing: G.J., J.R., Y.L.Z., S.Y., R.P., F.J.Z., A.J., F.D.A., X.S., X.C., Y.H. and K.O.

## Competing interests

The authors declare no competing interests.
