## [Peer Review File · Nature Communications]

Reviewers' Comments:

Reviewer #1:

Remarks to the Author:

The authors took a unique approach by developing a Tn7-based molecular barcoding strategy in *Ralstonia* to establish the barriers to infection, which is critical to determine the basis for population genetic effects throughout the infection process, such as the bottleneck effect of establishing in the host environment. This was an interesting study, and as far as I know, is a novel approach to the study of plant pathogens (similar methods have been done in animal models). In particular, I think the addition of a susceptible and resistant plant host and the different routes of infection make this manuscript's results noteworthy and have not been done in animal models.

I am a bit concerned about the beta diversity analyses performed. Although I understand the appeal of using the barcodes as "genetic diversity", they are still random synthetic barcodes that should not influence strain fitness and therefore the genetic diversity is really only informative as is to an extent. I have elaborated in more detail below as well as suggestions. I think the alpha diversity and population size estimates are appropriate. I also think most of the discussion that extends the barcoding to conceptualize the genetic diversity of a pathogen and extending this model to the natural world makes sense.

Major comments:

-For the ANOVA models (e.g. Fig 3B), was an interaction term tested? It does not appear so and only the main effects are tested, but some analyses appear like there might be interactions. I would also expect given the drastic phenotypic differences between the two host genotypes that there would be an interaction. I wonder if a nested model may be more appropriate to look at the differences by infection route and culture nested within the infection route to compare the differences among host genotypes. This would allow you to (for example with Fig 3B), as whether there are differences by infection route and then the role that host genotype plays in each case.

-I wonder how informative the beta diversity analyses are. The barcodes are not real forms of genetic diversity and presumably are all nearly identical apart from the barcode. Therefore, the alpha diversity metric and population size estimates are probably more informative to understand the bottleneck. To elaborate on this:

-For the PCoA, the barcodes that arise to dominant the population are random (as seen in Fig 3A), and the differences in composition are not as informative as the alpha diversity indices. It does highlight the variability noted in Fig 3A, which may be a more informative way to summarize that data (which was communicated well in the discussion). Instead of a PERMANOVA, the betadisper function to assess variability among each treatment centroid would be more informative to highlight the increased variability among the barcodes that make it into the host as permeability decreases.

-For the population assembly analysis with the BNTI, this seems like an inappropriate analysis because the barcodes are not phylogenetically informative, which is the basis for this analysis. I'm not sure if there is an alternative that is quite the same, but there are other taxonomic models, such as Sloan's neutral model and Raup-Crick model, that would indicate deterministic and stochastic processes.

Minor comments:

Abstract/Discussion/Conclusion: I wouldn't refer to this as a novel barcoding approach as it has been done a couple of times with Tn7 (e.g. <https://doi.org/10.1038/s41559-020-1103-z> and [10.1007/s00239-023-10103-6](https://doi.org/10.1007/s00239-023-10103-6))

L71: the invasive propagules "that are" able...

L95: high resolution "manner"

L250: Do you mean the maximum posterior alignment method for muscle?

L420: The ref 82 is not based on barcoding, but rather a statistical model that is informed by genome-derived haplotypes.

L426: there is an extra e in this sentence.

Fig 2a: For the schematic, it might be informative to explain a bit more in the figure legend or supplement some of the details at the beginning (such as the sequence presented, details are missing like why are some sequences underlined, etc).

Fig 2b: units for the growth curve analysis – I assume that the biomass is from the growth curve analysis, but it is unclear as is from the figure legend or methods.

Fig 2c: The light color used for the 10 strain barcode analysis is hard to see. Might be better to darker the contrast a little.

Fig 3B: it looks like the axis is on a log scale. It would be helpful to indicate this in the figure legend and also add some values between zero and 100. As is it is difficult to interpret the lower values.

Fig 4 correlations, please as the rho value for the Spearman correlation. Also add somewhere (for example, figure legend) how the line was generated and presumably the CI intervals.

Fig 4 correlations, do the host genotypes respond differently?

Reviewer #2:

Remarks to the Author:

Dear Editor: I am glad to review an interesting manuscript on the invasion process of a bacterial plant pathogen to the host and the role played by natural selection or alternatively random drift. In the manuscript by Jiang and collaborators, the authors focus on studying the population bottlenecks that may suffer along the process of invading plant tissues. The document is well written and experiments are well designed, with respective controls and robust statistical analysis. The objectives are clearly stated, under a clearly explained hypothesis. Despite the great strengths of the job, I find some aspects that worry me. In simple words, the authors find obvious results, which is: if there are more barriers to the entry of bacteria to the plant host, there will be a bottleneck problem reducing the genetic diversity of the bacteria population involved. It is not clear to what extent random drift plays a larger role than selection forces. More importantly, the authors do not even discuss the role that plant cell wall-degrading enzymes (CWDE) may play during the invasion of plant tissues. *Ralstonia solanacearum* (Rs) harbors several CWDEs that are virulence determinants and contribute to its ability to cause wilt disease. In real life, Rs secretes CWDEs that help to overcome plant-tissue barriers reducing or lowering permeability, therefore allowing access to xylem vessels. Similarly, some or a few Type 3 secreted proteins (effectors) may be critical for the first steps during the infection process. Although this work has the merit of indicating the reductions in genetic diversity that the bacterium undergoes during infection, I would have liked to see a more complete work that includes the factors described above that would determine the infective success of this pathogen.

Some other comments:

- 1) As an internal control, authors evaluate the impact of DNA barcode introduction on fitness and virulence by doing experiments using a rich medium. There is no similar investigation using a minimal medium. Since the natural environment of Rs is soil and plant xylem where there are very few organic nutrients but rather water with some salts, the experiment should have included a minimum medium.
- 2) The word propagules is usually linked to plant structures that can give rise to a new individual organism, especially parts of a plant that serve as means of vegetative reproduction, so it seems it does not fit well when referring to bacteria. Please use another word.
- 3) It is not clear if the dominant strains (i.e. in IR experiments) have acquired spontaneous mutations. Rs is quite prone to mutations because its high content of mobile elements and

unstable genome. Have the authors fully sequenced the barcoded strains that were raised as dominant genotypes?

4) It would have been good to include an "advanced" genotype that harbors some enzyme(s) to degrade the plant cell wall or some T3SS effector as an alternative with higher fitness that can help rule out the participation of randomness in the invasion process. For sure, OE1-1 must have a group of CWDE and effectors, however, an advanced strain would be useful to study selection in this system. If it is not possible to include new experiments, at least the role of CWDE/effectors should be discussed profusely to take this investigation to more real ground.

5) Please define Hes, Hos, and Und in the legend of Figure 5.

Response to Reviewer Comments

Reviewer #1 (Remarks to the Author):

The authors took a unique approach by developing a Tn7-based molecular barcoding strategy in *Ralstonia* to establish the barriers to infection, which is critical to determine the basis for population genetic effects throughout the infection process, such as the bottleneck effect of establishing in the host environment. This was an interesting study, and as far as I know, is a novel approach to the study of plant pathogens (similar methods have been done in animal models). In particular, I think the addition of a susceptible and resistant plant host and the different routes of infection make this manuscript's results noteworthy and have not been done in animal models.

Response: Thank you for the positive assessment of our manuscript.

I am a bit concerned about the beta diversity analyses performed. Although I understand the appeal of using the barcodes as “genetic diversity”, they are still random synthetic barcodes that should not influence strain fitness and therefore the genetic diversity is really only informative as is to an extent. I have elaborated in more detail below as well as suggestions. I think the alpha diversity and population size estimates are appropriate. I also think most of the discussion that extends the barcoding to conceptualize the genetic diversity of a pathogen and extending this model to the natural world makes sense.

Response: Thanks for the constructive comments. We have carefully addressed these concerns below. In brief, although the ‘artificial’ effect of the barcodes on beta-diversity, these analyses were maintained as they provide additional details that complement the alpha-diversity plots and data presentation. For instance, it more clearly displays differences and similarities (using community distance metrics) of the invading pathogen populations across treatments. See below for details.

Major comments:

-For the ANOVA models (e.g. Fig 3B), was an interaction term tested? It does not appear so and only the main effects are tested, but some analyses appear like there might be interactions. I would also expect given the drastic phenotypic differences between the two host genotypes that there would be an interaction. I wonder if a nested model may be more appropriate to look at the differences by infection route and culture nested within the infection route to compare the differences among host genotypes. This would allow you to (for example with Fig 3B), as whether there are differences by infection route and then the role that host genotype plays in each case.

Response: Thanks for this suggestion. We have now tested the interaction effect of cultivars and infection routes using Two-way ANOVA. In brief, we found significant differences in bottleneck size between the two cultivars ($F_{66,1} = 79.65, P < 0.001$), and the three infection routes ($F_{66,2} = 56.30, P < 0.001$). The interaction between these two factors was also found to be significant ($F_{66,2} = 15.66, P < 0.001$) for bottleneck size. The manuscript was revised accordingly as shown below.

311 To determine the impact of plant barrier permeability on pathogen population size, we compared
 312 differences in bottleneck size (N_b) between the two tomato cultivars and the three infection
 313 treatments (Fig. 3B). Both cultivars and infection treatments contributed to variation in bottleneck
 314 size with a significant interaction effect (Cultivar: $F_{66,1} = 56.30$, $P < 0.001$; infection treatment: $F_{66,2}$
 315 $= 79.65$, $P < 0.001$; interaction: $F_{66,2} = 15.66$, $P < 0.001$, Two-way ANOVA, Fig. 3B). The pathogen
 316 populations in the IR treatment had the strongest bottleneck with N_b values of 4.96 and 1.60 in the
 317 R and S plants, respectively ($P < 0.001$, ANOVA with LSD-test, Fig. 3B and Fig. S4). The N_b
 318 median values in treatments WR and DXI were 10.31 and 28.57 times higher than those of IR in
 319 R plants (Resistant: $F_{33,2} = 29.52$, $P < 0.001$, LSD-test), and 121.75 and 128.65 times higher than
 320 those of IR in S plants (Susceptible: $F_{33,2} = 53.04$, $P < 0.001$, LSD-test, Fig. 3B).⁴

-I wonder how informative the beta diversity analyses are. The barcodes are not real forms of genetic diversity and presumably are all nearly identical apart from the barcode. Therefore, the alpha diversity metric and population size estimates are probably more informative to understand the bottlenecking. To elaborate on this:

Response: Thanks for pointing this out. The main reason to use a beta-diversity analysis here is to provide an overview of the distances between the pathogen populations (similar to 'communities') invading the plant tissues. We acknowledge that the barcodes were artificially added to the strains, however, the main point here is not to reflect barcode sequence differences but rather to account for distinct pathogen populations and their relative abundances – which collectively would reflect in similarities (or dissimilarities) in the invading pathogen population across treatments. To the best of our knowledge, beta-diversity analysis is the only way to calculate and display such differences.

-For the PCoA, the barcodes that arise to dominant the population are random (as seen in Fig 3A), and the differences in composition are not as informative as the alpha diversity indices. It does highlight the variability noted in Fig 3A, which may be a more informative way to summarize that data (which was communicated well in the discussion). Instead of a PERMANOVA, the betadisper function to assess variability among each treatment centroid would be more informative to highlight the increased variability among the barcodes that make it into the host as permeability decreases.

Response: Thanks for the suggestion. As mentioned above, we maintained the beta-diversity analysis and – in addition – we now included the betadisper analysis. In brief, we initially used PERMANOVA to test for treatment differences and betadisper to assess variability among each treatment centroid. Information added in the revised manuscript.

338 ■ **Plant-tissue permeability impact phytopathogen lineage composition**⁴
 339 The composition of the invading phytopathogen populations was influenced by both host type
 340 (susceptible vs resistant) and infection treatment (PERMANOVA: $R^2 = 0.03$, $F_{66,1} = 2.93$, $P < 0.014$
 341 for host type; $R^2 = 0.32$, $F_{66,2} = 16.91$, $P < 0.001$ for infection treatment; $R^2 = 0.03$, $F_{66,2} = 1.81$, P
 342 < 0.001 for interaction effect, Fig. S7). Infection treatment resulted in different lineage composition
 343 profiles in R (PERMANOVA: $R^2 = 0.27$, $F_{33,2} = 13.02$, $P < 0.001$, Fig. 4d) and S cultivars
 344 (PERMANOVA: $R^2 = 0.44$, $F_{33,2} = 6.16$, $P < 0.001$, Fig. 4e). The variability of invading
 345 phytopathogen populations differed across infection treatments (PERMDISP: $F_{33,2} = 196.29$, $P =$
 346 0.001 in resistant; PERMDISP: $F_{33,2} = 421.51$, $P = 0.001$ in susceptible, Fig. S7). The multivariate
 347 dispersion of *R. solanacearum* populations within plant tissues was influenced by the host type
 348 and the infection treatment with a significant interaction effect (Cultivar: $F_{66,1} = 7.07$, $P = 0.010$;
 349 infection treatment: $F_{66,2} = 734.94$, $P < 0.001$; interaction: $F_{66,2} = 20.27$, $P < 0.001$, Two-way
 350 ANOVA, Fig. S7b). These effects segregated small population sizes and low genetic diversity in
 351 the IR treatment from other treatments ($P < 0.001$, LSD-test, Fig. S7b). The significant correlation
 352 between the N_b value with lineage dispersion further confirmed this finding ($Rho = -0.74$, $R^2 = 0.86$,
 353 $F_{70,1} = 451.2$, $P < 0.001$, $N_{ba} = 46.09$, Fig. 4f and Fig. S6d).⁴

-For the population assembly analysis with the BNTI, this seems like an inappropriate

analysis because the barcodes are not phylogenetically informative, which is the basis for this analysis. I'm not sure if there is an alternative that is quite the same, but there are other taxonomic models, such as Sloan's neutral model and Raup-Crick model, that would indicate deterministic and stochastic processes.

Response: These analyses were revised accordingly. First, we revised the entire β NTI model analysis and redefined an index named Beta artificial taxon index (β ATI), which is calculated using a similar method as the β NTI. We have now explicitly considered the fact that sequence variations are attributes of the inserted random barcode sequences. In addition, we now provide an analysis of the data using the Sloan's null modeling approach, which corroborates with the results obtained via β ATI analysis. The entire text, figure panels, and Supplementary figures were revised accordingly. See main text for details.

Result section:

354 **Plant-tissue permeability impact the assembly processes structuring *R. solanacearum*** 355 **populations**

356 To determine the relative influence of assembly processes structuring *R. solanacearum*
357 populations during invasion, we used β ATI and RC_{bray} metrics (Fig. 5), and the goodness of fitting
358 of the Sloan's model (Fig. S8). First, we found invading populations of *R. solanacearum* to be
359 mostly structured by deterministic selection in treatment IR (reflecting a low permeability to
360 invasion – and also associated with greater population and genetic bottlenecks). On the contrary,
361 the relative influence of stochastic processes increased in treatments WR and DXI (reflecting a
362 greater permeability to invasion – and also associated with lower population and genetic
363 bottlenecks) (Fig. 5). In detail, the IR treatment resulted in a β ATI > 2 (Fig. 5a), which suggests the
364 predominance of variable selection, both in R and S plants (Fig. 5b). At the intermediate level of
365 permeability (treatment WR), the proportion of variable selection was found to be lower and
366 homogeneous selection (β ATI < 2) was the predominant assembly process. Last, when plant
367 barriers were highly permeable to invasion (treatment DXI), the population assembly was mostly
368 structured by stochastic processes ($|\beta$ ATI| values were < 2), further determined via RC_{bray} analysis
369 as undominated processes, in both S and R plants (Fig. 5b). Based on the Sloan's model, the
370 frequency of barcoded strains occurrence in the IR treatment (Susceptible: $R^2 = 0.07$, Resistant:
371 $R^2 = 0.01$) was less fitted to the model in comparison with WR (Susceptible: $R^2 = 0.53$, Resistant:
372 $R^2 = 0.49$) and DXI (Susceptible: $R^2 = 0.88$, Resistant: $R^2 = 0.81$) treatments (Fig. S8a). The
373 population bottlenecks (N_b) had a strong correlation with β ATI values, whereby a reduction in the
374 strength of the population bottleneck during invasion shifted the assembly processes from
375 predominantly deterministic to predominantly stochastic based on the established pathogen
376 invader populations (Fig. 5c). The goodness of fit of the Sloan's model (R^2) increased along with
377 the bottleneck size and plant tissue permeability, thus suggesting an increasing role of stochasticity
378 structuring the assembly of the phytopathogen populations (Fig. S8c)

Method section:

259 *Analysis of ecological processes structuring population assembly*

260 We used two distinct models to investigate the effects of the plant tissue permeability on the
261 ecological processes structuring phytopathogen populations *in planta*. First, the assembly
262 processes structuring *R. solanacearum* barcoded populations were quantified using the beta
263 nearest taxon index (β NTI) and the Bray-Curtis-based Raup-Crick metric (RC_{bray})^{50,51}. By
264 considering the differently barcoded populations as artificial 'species' and the 25 bp barcodes as
265 the 'variable regions of a house-keeping gene', we were able to determine the relative importance
266 of stochastic and deterministic processes structuring *R. solanacearum* populations in response to
267 plant-tissue permeability to invasion (Fig. 1B). We first generated a maximum likelihood
268 phylogenetic tree based on a multiple sequence alignment of distinct barcode DNAs using the
269 parallel perturbed probcon method in MUSCLE v5⁵². We then calculated the beta artificial taxon
270 index (β ATI) adapted to the β NTI metric under the "taxa.jables" null model with 999 permutations
271 using the *ses.MNTD* and *comdistn* functions in the *picante* package v1.8.1⁵³. Specifically, $|\beta$ ATI| >
272 2 indicates the predominance of deterministic processes, where β ATI > 2 and β ATI < -2 indicate
273 variable and homogeneous selection, respectively²⁰. When $|\beta$ ATI| values were < 2, RC_{bray} was
274 used to partition the relative influence of distinct stochastic processes: $RC_{bray} < -0.95$, $RC_{bray} >$
275 0.95 , and $|RC_{bray}| < 0.95$ indicate the predominance of homogenizing dispersal, dispersal limitation,
276 and undominated processes (drift), respectively⁵⁰. We also used the Sloan's model⁵⁴ to estimate
277 expectations in abundances and frequencies of *R. solanacearum* barcoded strains. In brief, the
278 Sloan model estimates the immigration rate (m) to the *in planta* populations by comparing the
279 barcode frequency across plants. The barcode distribution was fitted the Sloan model using the
280 *neutral.fit* function in the *MicEco* package v0.9.19⁵⁵. Calculations of 95% confidence intervals for
281 the Sloan model were used to detect whether the frequency of barcoded strains was above or
282 below the data distribution expectation. The parameter R^2 indicates the goodness of fit of the model.

Minor comments:

Abstract/Discussion/Conclusion: I wouldn't refer to this as a novel barcoding approach as it has been done a couple of times with Tn7 (e.g. <https://doi.org/10.1038/s41559-020-1103-z> and [10.1007/s00239-023-10103-6](https://doi.org/10.1007/s00239-023-10103-6))

Response: Corrected accordingly. The word 'novel' was replaced with 'unique' or removed in some sections. Abstract: Line 58; Discussion: Lines 392 and 395; Conclusion: Lines 465 and 473 in the revision.

L71: the invasive propagules "that are" able...

Response: Corrected accordingly. See Lines 79-80 in the revised text.

L95: high resolution "manner"

Response: Corrected accordingly. See Line 104 in the revised text.

L250: Do you mean the maximum posterior alignment method for muscle?

Response: No. We used the parallel perturbed probcon method for the sequence alignment of DNA barcodes in MUSCLE. Corrected accordingly.

266 of stochastic and deterministic processes structuring *R. solanacearum* populations in response to
267 plant-tissue permeability to invasion (Fig. 1B). We first generated a maximum likelihood
268 phylogenetic tree based on a multiple sequence alignment of distinct barcode DNAs using the
269 parallel perturbed probcon method in MUSCLE v5⁵². We then calculated the beta artificial taxon

L420: The ref 82 is not based on barcoding, but rather a statistical model that is informed by genome-derived haplotypes.

Response: This reference was removed in the revised version.

463 dynamics over time, for example, in response to environmental disturbances or across plant growth
464 stages⁶⁸. When used on populations with known pre-existing mutations, random barcoding can
465 aid the testing of the fitness of pathogens or microbial inoculants (e.g., strain-level variation of host-
466 associated microorganisms).⁴

L426: there is an extra e in this sentence.

Response: Correction made. See Line 472 in the revised text.

Fig 2a: For the schematic, it might be informative to explain a bit more in the figure legend or supplement some of the details at the beginning (such as the sequence presented, details are missing like why some sequences are underlined, etc).

Response: Additional text is now provided in the legend of Fig 2a.

Revised Figure:

Revised Legend:

661 **Figure 2. Strain-specific DNA barcoding as an approach to track strain-level population**
662 **dynamics of phytopathogens.** (a) General scheme of the random chromosomal barcoding
663 approach and the experimental design. First, we edited a 25-nucleotide random sequence (the
664 'barcode') and the adjacent marker of selection between the Tn7 arms on the integration plasmid.
665 The helper plasmid and Tn7 plasmid were then integrated at a constant, neutral position in the
666 chromosome of the *R. solanacearum* cells by inducing the transposase machinery. The fate of
667 each barcoded strain can then be tracked by sequencing; Tn7 insertion doesn't impact strain
668 fitness and virulence (i.e., no differences in growth and infection *in planta*), measured as (b)
669 maximum growth rate and biomass (c) of the 90 isogenic barcoded *R. solanacearum* strains and
670 their even mixture. The red dashed line and the intersection represent the mean and the 95% CI
671 of maximum growth rate and biomass of the ancestral wild-type and barcoded strains. The purple
672 and blue boxplots denote the ancestral wild-type strain and the equal mixture pool of the 90-
673 barcoded strains; (d) proportion of the barcoded individuals after *in vitro* growth for 24 h initially
674 mixed at equal concentrations with an increasing number of total barcoded strains per inoculum.
675 The x-axis represents the starting number of mixed-barcoded strains, and the y-axis represents
676 the proportion of each barcoded strain. The dashed gray lines indicate the relative mean proportion
677 of each barcoded strain in the inoculum; (e) survival curve of tomato plants inoculated with the
678 pathogen wild-type strain and the barcoded strain pool of *R. solanacearum*. *P* values > 0.05
679 indicate no fitness variation of barcoded strains on the pathogen virulence based on the log-rank
680 test of survival curves.^d

Fig 2b: units for the growth curve analysis – I assume that the biomass is from the growth curve analysis, but it is unclear as is from the figure legend or methods.

Response: Revised for clarity. Fig 2b was changed to Fig 2b and 2c for the growth and biomass respectively. In brief, units were added in the legend of Fig 2b and 2c and additional text is now provided in the Methods section (growth parameters were calculated using growth curve analysis with the *gcFitModel* function to fit growth data with the *grofit* R package). Please, see below:

Fig 2c: The light color used for the 10 strain barcode analysis is hard to see. Might be better to darker the contrast a little.

Response: Corrected accordingly. Fig 2c is now Fig 2d in the revised manuscript.

Fig 3B: it looks like the axis is on a log scale. It would be helpful to indicate this in the figure legend and also add some values between zero and 100. As is it is difficult to interpret the lower values.

Response: The Y-axis displays the values of bottleneck size (N_b) using a sqrt scale. In the revised text, the values '10' and '50' were included in the Y-axis between zero and 100.

Fig 4 correlations, please as the rho value for the Spearman correlation. Also add somewhere (for example, figure legend) how the line was generated and presumably the CI intervals.

Response: The rho values of the Spearman correlation were included in panels c and f. The lines along the fitting curve denote CI intervals. Information added in Figure and legend.

Revised Figure:

Revised Legend:

685 **Figure 4. Genetic diversity of *R. solanacearum*, cell lineage composition, and their**
686 **correlation with population bottlenecks are affected by plant-tissue permeability.** Genetic
687 diversity (Shannon index) of *R. solanacearum* populations infecting susceptible (a) and resistant
688 (b) tomato cultivars across treatments. The inoculation treatment reflects variation in permeability
689 imposed by plant physical barriers to invasion (IR, intact roots; WR, wounded roots; DXI, direct
690 xylem infection); (c) correlation between bottleneck size (N_b) and the genetic diversity of invader
691 *R. solanacearum* populations. Principal coordinates analysis (PCoA) based on Bray-Curtis
692 distances displays a significant difference in *R. solanacearum* lineage composition across
693 inoculation treatments in resistant (d) and susceptible (e) plants; (f) correlation between *R.*
694 *solanacearum* population bottleneck size (N_b) and cell lineage composition (the first principal
695 coordinate component in panel (d) after invasion. **Statistics are provided as inset panels (d and e,**
696 **PERMANOVA with 9,999 permutations).** In panel c and f, Rho represents the Spearman's rank
697 correlation coefficient; Line and intersection represent the mean and 95% CI of fitting curve of
698 asymptotic regression model. **R2: goodness of fitting.**

Fig 4 correlations, do the host genotypes respond differently?

Response: Thanks for this question. The fitting curves of host genotypes showed similar patterns but with differences in *Rho* (ρ) values as below. This figure is now included as Supplementary Figure 6.

Supplementary Fig. 6. Correlation between bottleneck size (N_b) and the diversity of invader *R. solanacearum* populations. Diversity included Shannon index (a), observed richness (b), Pielou's evenness (c), and multivariate community dispersion (d). The treatments represent distinct plant-tissue permeability to invasion (i.e., intact roots, IR; wounded roots, WR; direct xylem inoculation, DXI). *Rho* (ρ) represents the Spearman's rank correlation coefficient; Line and intersection represent the mean and 95% CI of fitting curve of asymptotic regression model.

Reviewer #2 (Remarks to the Author):

Dear Editor: I am glad to review an interesting manuscript on the invasion process of a bacterial plant pathogen to the host and the role played by natural selection or alternatively random drift. In the manuscript by Jiang and collaborators, the authors focus on studying the population bottlenecks that may suffer along the process of invading plant tissues. The document is well written and experiments are well designed, with respective controls and robust statistical analysis. The objectives are clearly stated, under a clearly explained hypothesis.

Response: Thank you for the positive assessment of our manuscript.

Despite the great strengths of the job, I find some aspects that worry me. In simple words, the authors find obvious results, which is: if there are more barriers to the entry of bacteria to the plant host, there will be a bottleneck problem reducing the genetic diversity of the bacteria population involved.

It is not clear to what extent random drift plays a larger role than selection forces.

Response: We have now used a combination of β ATI and RC_{bray} metrics – in addition to the Sloan's model – to provide a better view of these processes structuring the invading population assembly across treatments. In brief, we found the relative influence of selection and stochasticity to shift based on plant-tissue permeability. In particular, when plant-tissue permeability is high the invading population assembly was mostly determined by stochasticity. Further RC_{bray} analysis suggested this stochasticity to be mostly attributed to ecological drift (also termed in the model as 'undominated processes') – as opposed to homogenizing dispersal or dispersal limitation. On the other hand, when tissue permeability was low, there was a dominance of deterministic selection structuring the invading populations in both cultivars.

More importantly, the authors do not even discuss the role that plant cell wall-degrading enzymes (CWDE) may play during the invasion of plant tissues. *Ralstonia solanacearum* (Rs) harbors several CWDEs that are virulence determinants and contribute to its ability to cause wilt disease. In real life, Rs secretes CWDEs that help to overcome plant-tissue barriers reducing or lowering permeability, therefore allowing access to xylem vessels. Similarly, some or a few Type 3 secreted proteins (effectors) may be critical for the first steps during the infection process. Although this work has the merit of indicating the reductions in genetic diversity that the bacterium undergoes during infection, I would have liked to see a more complete work that includes the factors described above that would determine the infective success of this pathogen.

Response: We thank the reviewer for this suggestion. This information was added and contextualized in the revised version of the manuscript; see L115 in the introduction and L452 in the Discussion section.

Despite the key importance of these metabolisms of pathogenicity, the explicit inclusion of these factors into our study was not possible or even clearly conceivable. In addition, since we used isogenic lines of *R. solanacearum*, we assumed all barcoded populations to have a similar capacity to express these virulence metabolisms.

Some other comments:

1) As an internal control, authors evaluate the impact of DNA barcode introduction on fitness and virulence by doing experiments using a rich medium. There is no similar investigation using a minimal medium. Since the natural environment of Rs is soil and plant xylem where there are very few organic nutrients but rather water with some salts, the experiment should have included a minimum medium.

Response: Since the barcode sequence constitutes only an additional 25 bp in the entire chromosome, we did not expect it to have any significant effect on *R. solanacearum* populations' fitness and virulence. As such, this experiment was carried out as a proof-of-principle rather than an attempt to mimic the conditions in soil or in the plant xylem. We believe that any attempt to mimic the soil or plant xylem environment would lead to bias in our assessment. To address the reviewer's concern, we have now performed the same experiment in a minimum medium and no significant difference was observed, see below: (the minimum medium was supplemented with 20 mM of L-glutamate as the sole carbon source). The text was revised accordingly, Methods, Results, and Supplementary Fig. 1 in the revised manuscript.

2) Propagules are usually linked to plant structures that can give rise to a new individual organism, especially parts of a plant that serve as means of vegetative reproduction, so it seems it does not fit well when referring to bacteria. Please use another word.

Response: Revised accordingly throughout the entire manuscript.

3) It is not clear if the dominant strains (i.e. in IR experiments) have acquired spontaneous mutations. *R. solanacearum* is quite prone to mutations because of its high content of mobile elements and unstable genome. Have the authors fully sequenced the barcoded strains that were raised as dominant genotypes?

Response: No, we did not. That would require additional efforts and complex comparative genome analysis that would extend beyond the scope of the article. Besides, we believe that the timeframe of our study would not favor spontaneous mutations to become a driving factor of our findings. Additional support for this assertion is provided in the literature, see below:

(1) mutations in *R. solanacearum* associated with fitness gain *in planta* were described in the gene *efpR* after at least 200 generations of the pathogen. This relates to ~18 serial passage experiments in bean plants (Guidot et al. 2014, Multihost experimental evolution of the pathogen *Ralstonia solanacearum* unveils genes involved in adaptation to plants).

(2) The mutation rate of the strain GMI1000 (based on the Nal resistant method) was around 20 mutations/ 10^{10} cells/generation (Remigi et al 2014 Transient hypermutagenesis accelerates the evolution of legume endosymbionts following horizontal gene transfer).

4) It would have been good to include an "advanced" genotype that harbors some enzyme(s) to degrade the plant cell wall or some T3SS effector as an alternative with higher fitness that can help rule out the participation of randomness in the invasion process. For sure, OE1-1 must have a group of CWDE and effectors, however, an advanced strain would be useful to study selection in this system. If it is not possible to include new experiments, at least the role of CWDE/effectors should be discussed profusely to take this investigation to more real ground.

Response: We agree with the reviewer that an advanced genotype would be interesting to test. However, we decided not to include such an analysis as it would extend beyond the current scope of the paper. In addition, we believe this addition would not contribute to the main points demonstrated and modeled in our study.

To address this issue properly, we now added information about these metabolisms in the introduction and discussion sections. See lines 115 and 452.

5) Please define Hes, Hos, and Und in the legend of Figure 5.

Response: Corrected accordingly.

702 **Figure 5. Effects of plant-tissue permeability on the ecological processes structuring the**
703 **assembly of *R. solanacearum* populations.** (a) β ATI values of resistant and susceptible cultivars
704 in treatments IR (intact roots), WR (wounded roots), and DXI (direct xylem inoculation)
705 representing distinct levels of permeability to pathogen invasion; (b) relative contribution of
706 assembly processes structuring the phytopathogen populations *in planta*; Hos: homogenous
707 selection; Hes: heterogenous selection; Und: undominated processes; (c) relationships between
708 β ATI values and the population bottleneck (N_b) of the pathogen populations during invasion. Rho
709 represents the Spearman's rank correlation coefficient; Line and intersection represent the mean
710 and 95% CI of fitting curve of asymptotic regression model. R^2 : goodness of fitting. [†]

Reviewers' Comments:

Reviewer #1:

Remarks to the Author:

NCOMMS-23-13533A

Title: Effects of plant-tissue permeability on phytopathogen invasion and population bottlenecks

Thank you for your care for detail in updating this manuscript. I feel that my previous comments have been addressed. I quite enjoyed reading the new version of this manuscript overall, and I think that the updates the authors made from both reviewer suggestions improve the quality of the interpretation. Although I am still unsure about the inclusion of the BNTI method as stated previously with regard to lack of phylogenetic informativeness of barcodes, I think adding the Sloan's neutral model has quelled this inquiry given that they show very similar results.

I don't think this needs to be changed because the results are very evident, but generally, you wouldn't want to accompany a PERMANOVA if the permutational multivariate dispersion test is significant. This would be similar to heteroscedasticity with a Levene's test for a univariate ANOVA.

Minor:

L90-92 – This sentence is a little confusing. Perhaps reword? Are you trying to state that pathogen success is not only dependent their adaptations to overcome plant tissue barriers and innate immunity? I think it is the use of "in case" that is throwing me off.

L128: delete "this" in the phrase of this pathogen populations.

Fig S3 seems like a more convincing version of Fig 3A, for me at least. The heatmap version is kind of hard to follow.

Reviewer #2:

Remarks to the Author:

Dear Editor:

I am glad to read the new version of the manuscript NCOMMS-23-13533A. Certainly, the manuscript has improved because the authors have addressed most of the concerns and recommendations detailed in the first review. They explain changes made and present arguments against a few suggestions that seem to be inappropriate or out of the timeframe of this research. However, I would like to suggest authors add brief sentences to the manuscript to clarify two issues that they correctly explain in the rebuttal but seems that they missing in the text:

1) Authors used isogenic lines of *R. solanacearum* so all barcoded populations would have a similar capacity to express the virulence genes. This explanation must be presented in the context of the key importance of the pathogenicity determinants such as CWDE and T3SS effectors.

2) The timeframe of this study would not favor spontaneous mutations as endorsed by other reported studies (please detail references). In this way, authors rule out possible bias due to mutations.

Besides these suggestions, I suggest to accept this work for publication.

Best

Response to Reviewer Comments

Reviewer #1 (Remarks to the Author):

Thank you for your care for detail in updating this manuscript. I feel that my previous comments have been addressed. I quite enjoyed reading the new version of this manuscript overall, and I think that the updates the authors made from both reviewer suggestions improve the quality of the interpretation. Although I am still unsure about the inclusion of the BNTI method as stated previously with regard to lack of phylogenetic informativeness of barcodes, I think adding the Sloan's neutral model has quelled this inquiry given that they show very similar results.

Response: Thank you for the criticism and feedback provided. All of which greatly improved our manuscript's analytical quality and readability.

Also, based on this last feedback, we decided to include the following statement in the manuscript (see Line 314-Line 319):

It is also worth acknowledging the limitation of the beta nearest taxon index (β NTI) due to the artificial nature of the barcodes. In brief, the random assignment of barcodes does not align with the assumption of phylogenetic conservatism (often used in β NTI studies)⁵⁰. As such, we used a beta artificial taxon index (β ATI) to avoid confusion. Most importantly, our β ATI results corroborate with the output obtained from Sloan's model (a null model that does not include phylogenetic information)⁵³.

I don't think this needs to be changed because the results are very evident, but generally, you wouldn't want to accompany a PERMANOVA if the permutational multivariate dispersion test is significant. This would be similar to heteroscedasticity with a Levene's test for a univariate ANOVA.

Response: Thanks for this suggestion. We now removed the PERMANOVA, and maintained the permutational multivariate dispersion analysis.

Minor:

L90-92 – This sentence is a little confusing. Perhaps reword? Are you trying to state that pathogen success is not only dependent their adaptations to overcome plant tissue barriers and innate immunity? I think it is the use of “in case” that is throwing me off.

Response: Thanks for pointing this out. This sentence is rephrased and now it reads: “For instance, the success of pathogen invasions is not only dependent on their capacities to thrive in the rhizosphere but also on their abilities to overcome plant tissue barriers and the plant innate immunity.”

L128: delete “this” in the phrase of this pathogen populations.

Response: Corrected accordingly.

Fig S3 seems like a more convincing version of Fig 3A, for me at least. The heatmap version is kind of hard to follow.

Response: Thanks for pointing this out. We now inverted the placements of Figures S3 and 3A in the revised manuscript.

Reviewer #2 (Remarks to the Author):

Dear Editor:

I am glad to read the new version of the manuscript NCOMMS-23-13533A. Certainly, the manuscript has improved because the authors have addressed most of the concerns and recommendations detailed in the first review. They explain changes made and present arguments against a few suggestions that seem to be inappropriate or out of the timeframe of this research. However, I would like to suggest authors add brief sentences to the manuscript to clarify two issues that they correctly explain in the rebuttal but seems that they missing in the text:

1) Authors used isogenic lines of *R. solanacearum* so all barcoded populations would have a similar capacity to express the virulence genes. This explanation must be presented in the context of the key importance of the pathogenicity determinants such as CWDE and T3SS effectors.

Response: Thanks for pointing this out. We have added this information in the previous revised version of the manuscript (see L115).

Also, we now included one sentence in L153 to explicitly mention that the expression of pathogenicity determinants such as CWDE and T3SS effectors is not affected.

“Hence, results from the control treatments performed are consistent with no significant effects of the barcodes on the expression of key pathogenicity factors such as CDWEs and T3Es.”

2) The timeframe of this study would not favor spontaneous mutations as endorsed by other reported studies (please detail references). In this way, authors rule out possible bias due to mutations.

Response: We now included the following sentence in the Discussion section (see L251). *For example, the use of random barcode tracking of *E. coli* lineages exposed to subinhibitory antibiotic concentrations revealed that pre-existing mutations can be responsible for the fixation of particular lineages rather than adaptive mutations²⁴. We acknowledge that in our study, however, we cannot rule out whether pre-existing mutations could have influenced the estimates of population dynamics across our experimental treatments. Most importantly, *R. solanacearum* used in this study would not favor spontaneous mutations endorsed by quantified mutation rate³⁷ and the timeframe required for adaptive mutations (previously reported to only be observed after 13 in planta serial passages¹⁵).*